# Empowering Citizens through Perceptual Sensing of Urban Environmental and Health Data Following a Participative Citizen Science Approach

**DOI:** 10.3390/s19132940

**Published:** 2019-07-03

**Authors:** Manuel Ottaviano, María Eugenia Beltrán-Jaunsarás, José Gabriel Teriús-Padrón, Rebeca I. García-Betances, Sergio González-Martínez, Gloria Cea, Cecilia Vera, María Fernanda Cabrera-Umpiérrez, María Teresa Arredondo Waldmeyer

**Affiliations:** LifeSTech, Department of Photonics and Bioengineering, Escuela Técnica Superior de Ingenieros de Telecomunicación, Universidad Politécnica de Madrid, Avenida Complutense n° 30, Ciudad Universitaria, 28040 Madrid, Spain

**Keywords:** citizen science, pollution, public health, environmental sensors, sustainable lifestyle, green behaviour, user empowerment

## Abstract

The growth of the urban population together with a high concentration of air pollution have important health impacts on citizens who are exposed to them, causing serious risks of the development and evolution of different chronic diseases. This paper presents the design and development of a novel participatory citizen science-based application and data ecosystem model. These developments are imperative and scientifically designed to gather and process perceptual sensing of urban, environmental, and health data. This data acquisition approach allows citizens to gather and generate environment- and health-related data through mobile devices. The sum of all citizens’ data will continuously enrich and increase the volumes of data coming from the city sensors and sources across geographical locations. These scientifically generated data, coupled with data from the city sensors and sources, will enable specialized predictive analytic solutions to empower citizens with urban, environmental, and health recommendations, while enabling new data-driven policies. Although it is difficult for citizens to relate their personal behaviour to large-scale problems such as climate change, pollution, or public health, the developed ecosystem provides the necessary tools to enable a greener and healthier lifestyle, improve quality of life, and contribute towards a more sustainable local environment.

## 1. Introduction

The last century has experienced a very intense urbanization transition. It is foreseen that in 2050, 68% of the population will live in the cities. The growing of urban populations is manly driven by social and economic opportunities and the extended availability of services such as health, education, or public transportation [1]. Understanding the key trends in urbanization is crucial to understand the challenges in meeting the needs of their growing urban populations. 

Nowadays, cities generate 75% of the carbon emission, which means the urban population is highly exposed to pollutants and vulnerable to climate changes [2]. The World Health Organization (WHO) estimates that 63% of global mortality (about 36 million deaths per year) is the result of non-communicable diseases (e.g., cardiovascular disease, cancer, diabetes, and lung disease). A large proportion of these are attributable to the increasing exposure to air pollution concentration. According to the WHO, among all cities in the world, only 20% of the urban population lives in areas that comply with WHO air quality guideline levels regarding particulate matter (PM2.5) [3]. Thus, a high proportion of citizens are exposed to high levels of pollution, putting their health and development at serious risk in the long term [4]. This situation has driven to consider as a priority the need to create tools, guidelines, and recommendations to enable healthier, climate-friendly, and resilient lifestyles within urban environments [5]. Open data initiatives have been launched to provide broad access to data from the public sector, business, and science. In this regard, cities can leverage on digital and Internet of Things (IoT)-enabled technologies to provide actionable insights that help improve the health and wellbeing of citizens. Nowadays, digital technologies integrating sensors from homes, smartphones, wearables, and IoT devices are coming to the market at affordable prices for end users [6]. These technical trends allow citizens to gather and share information about themselves and the environment that surrounds them, known as citizen science. Furthermore, current technological resources represent an opportunity to assess the health status at the population and individual level, as well as identify the barriers and stressors that jeopardize citizens’ health and offer routes to develop new health policies [7].

In this context, the European funded project PULSE (Participatory Urban Living for Sustainable Environments) [8] aims to develop a public health observatory, related policies, and new procedures based on real evidence using data from different sources. Information from the public health systems, remote and fixed environmental sensors, and citizens’ mobile devices is collected in seven cities (Paris, Singapore, Birmingham, Barcelona, New York, Pavia, and Keelung) through a platform architecture and mobile application. Furthermore, the collected data are used to nourish the spatio-temporal and risk stratification models based on modifiable and non-modifiable asthma and type 2 diabetes (T2D) risk factors in each city, including behavioural, environmental, and social risks. The principles of citizens in science are integrated to enable public participation in scientific discovery, monitoring, and experimentation for improving and integrating the data generated into improved health and environmental policies. Following this approach, citizens become active in gathering, sharing, and understanding the data on the local environment using the PulsAir mobile application. 

This paper presents the design and development process of the PulsAir mobile application and data ecosystem model, which enable large quantities of public data collection and processing for striving on the yield potential of scientific research regarding the predictive environmental-health domain, sustainable healthier and greener oriented behaviours in urban environments, and new public policy tools for sustainable urban designs. The combination of the PulsAir application and the data ecosystem is conceived as a novel data-driven technological development designed to overcome information gaps, enabling the fusion of environmental and health data in cities, especially in local areas. Conventional approaches to air quality monitoring in cities are based on networks of static and mobile measurement stations, which are usually sparse and do not allow capturing tempo-spatial heterogeneity or identify pollution hotspots, contributing to the lack of information on health impact at the local level and on polluted areas [9]. Citizens consider that pollution is a main problem with a direct effect on their daily life and health requiring more actions to protect the environment [10]. The main contributions comprise the following: (1) the design and development of the PulsAir gamification-based application; and (2) the data production and consumption ecosystem and backend that integrates all data sources and allows data fusion, in order to link environmental data under a health-focused approach.

Through PulsAir, citizens provide information from the environment, position, or mobility, as well as data from activity sensors (e.g., Fitbit). Also, a set of questionnaires are triggered gathering scientific oriented information regarding environment and health aspects of the citizens. 

The data-driven ecosystem empowers citizens by enabling the data gathered processing to assess the citizens in terms of health risks (i.e., risk of the onset of type 2 diabetes, asthma, and cardiovascular disease (CVD)). The bidirectional path between the application and the data ecosystem allows generating a direct connection between the city and its inhabitants, empowering them to take actions towards improved and healthier behaviours. 

Besides citizen engagement and data processing, the data ecosystem and PulsAir application have been developed, integrating models that support the citizen science data generation and processing, including the fact of acquiring or vetting collected data. The data monitoring process links with the scientific environmental-health research goals through questionnaires that are related to scientifically validated protocols that trigger the tracking of physical activities and environmental pollution information.

The paper is organized as follows. Section 2 presents the materials used and the methods followed to design and develop the PulsAir application. Section 3 presents the results including the user needs and requirements collected from previous research and focus groups with key stakeholders, the prototype designs and gamification strategy followed, and the final application development, providing information of the workflow and the big data architecture. Finally, Section 4 presents the discussion and conclusions, as well as future research directions including evaluation of the application within the seven pilot cities.

## 2. Materials and Methods

The solution presented in this paper includes the conceptual design and development of a novel participatory citizen science-based mobile application used to empower the citizens through perceptual sensing of urban environmental and health data. This aims at promoting a green and healthy lifestyle increasing citizens quality of life. To accomplish this, the following materials and methods were applied.

### 2.1. Materials

Considering that the application should be able to collect data through urban environment sensing and should be available for iOS [11] and Android [12] smartphones to reach as many users as possible, the following materials were used:IONIC framework [13]: an open-source framework for the development of mobile and desktop application in different operating systems, based on HTML5 [14], CSS3 [15], JavaScript [16], and TypeScript [17]. The use of this framework allowed the development of the application for both Android and iOS devices at the same time.Cordova plugins: open-source development tool used to access to the native system features of the different platforms such as GPS or accelerometer [18]. This technology, also based on HTML5, CSS3, JavaScript, and TypeScript, was used to develop the smartphone native featuring for both Android and iOS devices through the IONIC development framework.AirVisual API: is a free tool that collects data from the largest network of ground-based sensors worldwide, providing air quality information. Air quality data based on the user GPS location from the smartphone were gathered using this API [19].Smartphones: these are the devices (i.e., iPhone and Samsung) used to validate the correct development of the application in both operating systems, iOS and Android. The minimum requirements of these devices are GPS, Bluetooth, and 4G connectivity in order to perform all the application functionalities. These devices are also the tool for citizen science data capture.City and Citizen Science Data: the participatory data collection approach, enabled through smartphones (by sending specific information provided by the user), wearable activity tracker devices, fixed and mobile environmental city stations, and satellite data.

### 2.2. Methods

In order to conceive the conceptual design and development of the participatory citizen science-based mobile application, the following methodologies were undertaken for the application system design and the big data-oriented architecture design.

#### 2.2.1. Participation Criteria

In order to recruit participants for the system validation, municipal leaders, community leaders, non-governmental organisations (NGOs), and local businesses were mobilized, and all parties were invited to use the application. In addition, at the city level, and in order to guarantee the sample size needed for the study, primary health care centres, community health workgroups, asthma and T2D patient’s associations, the public health agency, and social media were contacted.

The method considered a heterogeneous and balanced sample including age, gender, educational level, immigration status, and neighbourhood characteristics. Specifically, participants should meet the following inclusion criteria:Older than 18 years.Living in the urban areas covered by the pilot.Agreeing to participate in the project by signing a consent form explaining the different aspects of the study.Agreeing to wear an activity tracker.Owning an Android or IOS smartphone.

#### 2.2.2. Application System Design

The system was designed following a user-centred approach based on the goal-oriented design (GOD) methodology [20], in which end users and stakeholders guide the process, as they will ultimately validate the final product. This application was co-designed with public health organizations (PHOs), defining five different phases that are summarized as follows:First phase: definition of the user needs and focus.In this phase, research regarding the effects of environmental pollution on the quality of life of people, as well as the effects of an unhealthy lifestyle, was carried out in order to define the metrics for a multidimensional evaluation [21,22]. Furthermore, a benchmark assessment was conducted to identify applications and solutions that raise awareness among users of the effects of environmental pollution on health. Gamification techniques were analysed in order to define, in the second phase, the gamification strategy, to enhance users’ motivation [23]. Finally, focus groups were conducted with PHOs and users to define their needs and requirements to improve citizens’ quality of life.Second phase: creation of a non-functional prototype and gamification strategy.The first version of the system is a non-functional prototype that was built as a set of mock-ups that described the appearance and functionalities of the system. Mock-ups were revised and improved later considering the feedback received from end users. This approach allows to simulate/anticipate the possible user experience through two activities: (1) the definition of the most relevant screens of the application to specify the minimum viable set of functionalities; (2) the definition of the user interaction, which describes the workflow of the screens and its elements to perform a specific task. The non-functional prototype was conceived together with a gamification strategy. The gamification strategy defined is based on the application of game-design elements and game principles in non-gamified contexts [24] in order to produce positive effects on individuals [25]. Concretely, the main goal of the gamification process was to get a positive behaviour change regarding healthy and green habits.Third phase: development of the prototype.In this phase, the different elements of the application were developed, based on the feedback received of the non-functional prototype from phase two. These elements are as follows: (1) interfaces, from home screen to the different modules of the application; (2) communication and interaction with external services such as activity tracker devices and APIs; (3) application workflow, which includes trigger activities, feedback to user, educational content, monitor activities, gamification strategy, and so on. The development of the prototype followed a cross platform application approach, supporting Android and iOS devices, in order to cover all kinds of smartphone users.

#### 2.2.3. Consumption and Production Data Flow Process

A data backend ecosystem was developed to support the interoperable data flow to enable a big data oriented approach that generates valuable services. The ecosystem allows continuous interaction with the mobile application. This enables participative data gathering, interoperability among different sources, analysis and modelling over the widest array of information, interventions support, and data production to be consumed by users through the application.

In turn, the design and development of the mobile application integrates a data flow process that allows a continuous real time interaction and flow between the data production and consumption processes. To do so, four steps were followed defining four overlapping data contextual layers, as presented in Figure 1. These data layers were mirrored to the backend infrastructure for data processing and visualization of processed data and consumed through the mobile application, enabling real-time information delivery.

The four steps for data production and consumption flow comprise the following:Citizen science data gathering: this allows data gathering from the world of IoT and wearable devices that provide health status, presence, and use of a specific service. This includes a data acquisition phase binding the data streams to a common representational scheme to prepare the data for the sensor fusion and to counter some conditions with some pre-processing and elimination procedures (avoiding overloading of data), thus harvesting the data from the sensor units, tagging them with location and time information and storing data in real time. This supports data production and leverages data science.

The collected information enables the Users’ Context and comprises sensing and capturing of perceptual data regarding health, lifestyle, and habits behaviour (e.g., exercise, food, mobility). These data are linked with data coming from the Environmental Context.

Data sampling and collection: this process gathers environmental data from public data sources; mainly air quality data coming from satellites, fixed and mobile stations, as well as through the city sensors and citizen science mobile devices (smartphones and wearables). The collected information enables data production to be handled by the Environmental Context within the data ecosystem.Data set fusion: this process brings together data sampling and collection with high-quality pre-existing datasets (i.e., environment and health) to provide a comprehensive view of the environmental domain (e.g., risks, potential exposure) and integration with health information (e.g., asthma related symptoms). The collected information enables the Big Data Context, which is the operating environment that allows data fusion and spatio-temporal modelling in order to provide, for example, health risk estimates or pollution estimated exposure real-time information. This data layer also provides the urban data lake (including cloud-based servers), which stores the data and applies interpolation models to allow an interactive data consumption process, providing valuable insights and data to enable information-oriented services.Data interaction: this supports interactive data consumption through the mobile context and application. The collected information enables the Mobile Context. It provides a tool that hides the complexity of the data flows and provides the information in one coherent integrated visualization. It is a robust, user-friendly, and appealing interactive data layer that aims at providing data value and empowerment to citizens.

#### 2.2.4. Pilots Plan

The pilots plan comprises the following phases:Stage 1, training: municipal leaders, community leaders, NGOs, and local businesses received training, covering the objectives of the study and the PulsAir application functioning, in order to execute the recruitment of users.Stage 2, deployment: included the recruitment of users, the distribution of the wearable trackers, installation of PulsAir app in the smartphones, sign up with individual user access codes, and synchronization of the wearable trackers with PulsAir app.Stage 3, follow-up: during this phase, interactions with the pilot organisations are being maintained in order to perform refinements on the application according to users’ feedbacks and cultural requirements of each pilot city (e.g., language, educative systems, ethnic groups, religions, and metric systems among others).Stage 4, evaluation: includes the technical validation (e.g., performance evaluation, safety, and reliability assessment) and the assessment of data quality, user experience and usability, as well as social acceptance and effectiveness of the PulsAir application. The evaluation criteria includes the following measures: (a) social acceptance and effectiveness determined taking into account the working conditions (e.g., citizen science environment), attitude towards the outcome, reciprocity, and performance of the tools; (b) usability, which measures the effectiveness efficiency, learnability, robustness, and utility of the application and ecosystem; and (c) user experience taking into account embodiment, emotion, feeling of trust, and satisfaction. This phase is currently being implemented in the different cities involved in PULSE project deployment.

## 3. Results

The main results comprise the development of the interactive PulsAir application and the data ecosystem. The PulsAir was designed and developed integrating game-design elements, and feedback received from end users regarding requirements and needs for interface designs, functionalities, and workflow. The data production and consumption flow links the application with the backend processing, enabling a data ecosystem supported by four contextual data layers: the Mobile Context, the User Context, the Environmental Context, and the overall Big Data Context. In the following sections, a detailed description of these elements is presented.

### 3.1. User Needs and Requirements Driving the Citizen Science-Based Application

Users’ needs and requirements were collected during the first phase. Needs and requirements were collected through qualitative research techniques (e.g., focus groups), which took place in the different cities where the project is being piloted. The results from conducted research comprise the following key requirements:Mobile information and generation of higher density of data: there is a need to use heterogeneous mobile technologies (different brands and operating systems), commercial health trackers (e.g., Fitbit, Garmin), and the integration of new fixed IoT-based sensors in cities, thus engaging citizens for integrating citizen science. This is driven by the need for increasing the continuous gathering of quantitative and dynamic health and environmental perceptual data, in order to generate a higher density of health data and air quality measurements to provide a more accurate spatial and temporal data for health and environmental risk prediction and estimation.Privacy: the need of privacy requirements that ensure the system compliance with the General Data Protection Regulation (GDPR) to grant full control of the shared information to the users and a secure process of data gathering that can confer anonymization of the information.Air quality status and information: citizens reported the need to receive information and increase the awareness of pollution levels and exposure in the area where they live and perform activities. Also, it was reported as crucial to understand the pollutants and the status of the air pollution in the city. The city can leverage on the community’s air quality stations and on other networks of environmental devices available in the market that can be installed at home in balconies, gardens, rooftops, and so on. Furthermore, users would like to be able to see contributions to improve environmental actions in the city, and reported the need of having prospective routes, with alternative low polluted paths, to reach a destination in the mapping.Health-related assessment and recommendations: users reported the need to receive tailored feedback and information that will show areas and parameters that indicate health risk for their health situation. Also, recommendations, guidelines, and information that will impact on their health-oriented behavioural habits; showing somehow the contribution of the actions regarding health and community improvements.An appealing, easy to use, and robust application: users reported the need for an easy to use interface that will provide an enjoyable application. This meant to have a user-friendly and intuitive setting that could easily manage processes, data sharing, and participation with robust real-time processing and feedback.

### 3.2. Non-Functional Prototype and Gamification Approach

A team of user interaction experts, bioengineers, and graphical designers worked together to define a first approach of the user interaction solution. To do so, a first non-functional prototype (mock-up) was sketched based on the requirements gathered and iteratively validated with public health organizations through the pilot sites. The result of this phase was the characterization and design of the gamified application idea that visualizes interactive data and supports data consumption, based on citizens’ needs. The first activity was to consolidate a home page that will be able to show requirements, which were split as a twofold macro-functionality; one comprising requirements related to the users’ profile (“Me”) and the other with requirements related to the city in which the users are living (“My City”). For this purpose, the development team came up with two versions of the user interface (Figure 2). 

When requirements and feedback from the first iteration were integrated, a first version of the home page was consolidated, defining the specific functionalities for the citizens and city pages. 

Figure 2a shows the first design of the home screen that was modified later, as shown in Figure 2b, according to integrated feedback and detection of new users’ needs. This process aimed at finding a robust and appealing home screen that reflects the concept and rationale of the application. Changes included a new avatar and look and feel backgrounds, which were co-designed with users to provide a more city-oriented context. Figure 2c,d show the initial designs of the air quality information sections of “My city” and the Leaderboard of the application.

Furthermore, the design of the application integrates game-design elements in order to generate empowerment and motivation to users. To boost engagement and motivation, a live view of personal targets, benchmarks and/or achievements, and green and healthier behavioural values are provided in the home screen. This approach also enables clear goals and inspires more focused and efficient engaged citizens (e.g., higher scoring in using non polluted routes, higher scoring sharing air quality data) with a real impact on targeted activities. The following gamification mechanics, dynamics, and aesthetics were selected and integrated in the application for this purpose:Avatar: the visual representation of the user’s status in terms of health and green habits; it is the main character of the application. With this element, it is expected that users have a much higher chance of developing personal ownership.Points: used as status indicators and to reward users for their different behaviours or objectives achieved in the application, such as use of wearable activity tracker devices or filling in questionnaires and getting extra points depending on the answer, oriented to promote or to follow healthy and green habits checked with the data collected. Points are used to enable a reward mechanism that drives motivational and engagement behaviours through citizen science.Levels: Five levels were defined within the gamification strategy. The Level-up is achieved through the accumulation of points from the activities completed by the user. The levels represent the air quality conditions of the city in which the Avatar lives. User’s interactions with the application and accumulated points improve the conditions of the city. Level 1 is a highly polluted city, and Level 5 represents a clean and sustainable city.Rewards: tangible recompense (either physically or virtually) for the achievement of an objective through an action or series of actions proposed.Leaderboard: an ordered list with the user’s score (achieved points). Two categories are implemented, per city and worldwide.

### 3.3. The System Components

The overall system proposed comprises two main components: the PulsAir mobile application and the platform or processing data ecosystem. Design iterations are transversal to the system components development, allowing an incremental buildout progress. In the following sections, the system components, which naturally emerged during the iterative and co-creation design processes, are described in detail.

#### 3.3.1. PulsAir Mobile Application

The mobile application was developed enabling perception and sensing to be monitored and evolve over time. It collects and integrates environmental data and health information including patterns, demographics, and health and weather statistics, in order to provide citizens and authorities with a decision-making tool. The application was conceived as an engine to support valuable data interaction, data consumption, data capturing, and data sharing. The use of wearables linked to the application enables participatory sensing by using these devices as tools to collect and share health and air quality data obtained from personal sensing units. The application also gathers and provides personalised information by scheduling different type of questionnaires related to health and air quality, educational contents, and feedback to the users; the contents are personalized according to the usage of the app, the answered questions, and the profile of the user. Table 1 presents a list of the selected questionnaires.

The mobile application is composed of two main modules:“Me” module contains information about the citizen. It gathers all the specific information regarding the user, from questionnaires and from the wearable devices (e.g., Fitbit band). It also provides specific educational content to promote healthy lifestyles and feedback about the health risks using state of the art models for cardiovascular risk (Framingham, [30]), TD2M (FindRisk, [34]), and asthma (Verlato, [35]). The “Me” module shows the daily physical activity (steps) of the user and historical data of total activity from the previous days. The wearable device information is integrated in the app. The user can also access the “Health tips” to learn how to improve lifestyle based on the results of the answered questionnaires.“My city” module contains all the data related to the city and neighbourhood of the user. Through this module, the user has access to a page that summarizes the status of the city (i.e., pollution levels), specific contents about the city, the pollutants, and possible more sustainable mobility and household habits that can contribute to the reduction of polluted emissions. The module provides a detailed map to show pollution levels in different areas of the city. In addition, it allows the user to tag or input data through questionnaires to inform about the status of the neighbourhood. The information is personalized for each user, aiming at empowering the citizens with personal tools that estimate and manage their pollution exposure, and thus take healthier actions (i.e., less polluted routes when walking).

##### Mobile Application Interface

Figure 3 shows the final interface design of the application, specifically the following main elements:

Homepage, final design follows a card layout style with a simple and clean background. The cards give access to the different modules of the application, as shown in Figure 3a. From top to bottom, the cards show the following elements:“Me” card, shows the information about the number of questionnaires completed and the risk of developing cardiovascular disease, type 2 diabetes, or asthma, based on the results of the questionnaires completed.“My City” card, presents a summary of the air quality and weather conditions at the current location of the user.“Weekly Activity” card, presents the summary of the activities performed in terms of calories burnt, steps, and kilometres travelled. This information is collected through the activity tracker device.

The “Me” module screen shows the information gathered from the users; the wearable data and the health risks (Figure 3b). The menu, at the bottom of the screen, is used to access the answered questionnaires and suggested educational content and recommendations.The “My City” module shows the air quality and weather information about the city and the current user’s GPS position (Figure 3c). The menu, at the button on the screen, gives access to a pollution map and content related to air pollution.The usage of the application triggers the gamification engine, allowing users to earn points to level up through the five levels available (Figure 3d). The application starts with a sad avatar living in a dirty city; if the users perform the suggested activity within the application (i.e., questionnaire, educational contents, changes in habits), they will be taken to the next level until they reach the final level that shows a happy avatar living in a clean city (Figure 3e). The Leaderboard screen shows the user’s collected points (Figure 3f) in order to boost competitions among the citizens.

##### Mobile Application Workflow

The application workflow was conceived as a bidirectional interactive path. On one hand, the user shares gathered data with the platform, from physical activity measurements through wearable activity trackers and by answering the questionnaires. On the other hand, users receive back general or customized interventions depending on their profile. This intervention is composed of educational health and environmental guidelines or content, health risk awareness, geo-localized air pollution exposure and data, as well as gamification-oriented information and rewards based on their performance. This intervention aims at making data relevant to citizens to support greener and healthier sustainable behaviours. The process was defined using an incremental strategy, which comprises the following steps:Monitor: the application tracks GPS position and integrates data of wellness sensor (e.g., Fitbit) and air quality. From time to time, some questionnaire are triggered to the users. The application also tracks the degree of use of the application.Assess: the gathered information is used to assess the citizens in terms of health risks (risk of the onset of T2D, asthma, and cardiovascular disease).Personalize: according to the assessment, the application triggers specific contents and recommendations to the users. The application can identify healthy or green behaviour that needs to be improved and prepare a set of tips to help the user to be aware and understand the importance of modifying a specific behaviour.Act: the application promotes specific healthy habits by proposing a specific goal to the users (e.g., 8000 daily steps). According to this goal, the application creates a follow-up procedure to check if the goal is achieved in the proposed time (e.g., one day).Support and reward: when the user finish the proposed activity, the application triggers a mechanism of reward and supports the user to be motivated.

Figure 4 shows an example of the workflow of a specific case: health risk.

#### 3.3.2. Data Acquisition

The proposed approach gathers multilevel data coming from heterogeneous sources represented by structured or unstructured data, including citizen science. The data acquisition process is supported by both the mobile application and the data ecosystem, which have an embedded functional model that acquires scientifically sound information to further support predictive outputs and changes of environmental and health risks that could affect citizens and support policy making. The system continuously monitors pollution, as well as health and citizens’ data through data collection protocols and questionnaires (Figure 4) to provide evidence of the importance of detecting long-term environmental-health risks and ecosystem changes. Table 2 summarizes the types of information that are gathered in each data flow step (Section 2.2.2) within the proposed technical context.

The proposed approach gives the opportunity to create a data science approach to understand the relationship of the urban environment with the health of the citizens. The combination of different contextual data (i.e., mobile, user, environmental, and big data) generates new opportunities to merge different qualitative and quantitative methods, subjective and objective data, as well as sensitive and aggregated data to build new preventive strategies based on citizens’ needs.

#### 3.3.3. Big Data-Oriented Ecosystem

The big data-oriented ecosystem integrates the heterogeneous datasets and resources in a data lake, the urban data lake, under a big data architecture paradigm that allows the production and consumption of data flows. The ecosystem is the operational and transactional backend that supports all resources, open source toolsets, and operational data for developing and packaging services that support the modelling and real-time data provision/consumption from the mobile application. The data ecosystem acts as a backend that houses different physical nodes servers where information is stored, processed, and retrieved. This backend facilitates the delivery of services through the provision of distributed technologies, integration of standards, further scalability for science growth, and resource component management based in a distributed computing paradigm [36].

The data ecosystem responds to four levels of applications or contexts of information where data are gathered and processed: the mobile context (data interaction gathering), user context (citizen science data gathering), environmental context (data sampling and collection), and big data context (data fusion). These contexts provide different interlinked spaces, which ensure the integration of data collection and processing protocols needed to adhere to scientific standards, especially for publicly collected data. Figure 5 shows a diagram of the data ecosystem.

##### Mobile Context

This represents the data layer that supports the data consumption and interaction through the application. It provides a tool that hides the complexity of the data flows and provides the information in one coherent integrated visualization layer. This layer allows the user to capture, gather, and share personal data from the socio-demographic, health status, neighbourhood perception, geolocation, quality of life, and so on. Furthermore, the user can share the activity data of a health tracker device (e.g., Fitbit) and the GPS position. The application continuously displays information to the user, including health risks (e.g., asthma, cardiovascular, and type 2 diabetes risks) [37], feedback information, sensing services from the wearable activity tracker (e.g., steps, calories burnt, and kilometres walked), current location, and current pollution exposure values, among others. It is based on the following software artefacts able to run in different the different smartphones brand types (iOS and Android):Graphical User Interfaces (GUI): they represent all the screens as described in Section 3.3.1. They refer to four main activities for the users: the Citizen module that refers to the individual information and ways to promote healthy habits. The City module contains all the screens related to the health, green behavior, and pollution maps. “Preferences” contains all the screens for user’s preferences, and the “Home” page is the initial interaction flow.Intervention Engine (IE): this is a mechanism to provide the intervention in the application. It is based on a scheduler that executes a questionnaire engine able to process the intervention by proposing a set of activities to the user to promote green and healthy habits.Feedback Engine (FE): this is the logic that triggers the proper feedback to the users. There are multi modalities of feedbacks depending on the context and nature of the data. Feedback can be in form of charts (e.g., evolution of the daily steps from the health tracker), educational content (in form of texts, infographics, web pages, or videos), personalized feedbacks (messages build based on rules of a decision support system), or notifications or reminders that are triggered to push users to perform an activity.Sensing Service (SE): the artefact abstracts the complexity of the sensor connection and provides access to the internal sensor of the mobile phone (currently the GPS), sensors that are available on the WiFi network of the users, and third-party services.Synchronization Service (SS): this service communicates with the big data context to transfer the citizen data to the citizen database; the position of the user is then used (if available) to assess the environmental context and enrich the feedback to the users (e.g., pollution maps).App Database (AD): the service offers full data persistence inside the smartphone. The database is securely encrypted and stored in a non-accessible area of the smartphone.

##### User Context

This represents the world of IoT devices and wearable devices that can be used to get information about health status, presence, geolocation, and trajectories followed in space and time by single citizens using the application or a specific service. Users can decide to connect a specific device (e.g., Fitbit) and share the activity data with the platform. This research considered the use of a health tracker (i.e., Fitbit), but the types of devices can be extended to other health devices like holters, blood pressure monitors, and so on. This architectural layer allows citizen science data gathering that is uploaded in the cloud infrastructure.

##### Environmental Context 

This provides relevant information on the environment in the city area where the user is located. Data are captured from public information measurement nodes or infrastructure; mainly produced by the city sensors’ network (i.e., fixed or mobile sensors) or satellites (e.g., satellite optical or hyperspectral images). Additionally, these data are coupled with public services data (e.g., public health data) and resources, such as transportation, police, or other related public services aiming at supporting mobility.

Furthermore, this layer supports the data sampling and collection process. The data obtained comes from public access sources, which allows the most relevant information to be integrated in a single system for the user, and thus uploaded to the cloud infrastructure. The most relevant data gathered within the environmental and public services context comprise the following:Air Quality (AQ): Air quality data of the area where the user is located, coming from fixed and mobile sensors or measurement stations deployed throughout the city or satellites, among others.Weather: updated information of the weather in the user’s location.Mobility: public transport stops, pedestrian routes, bike lanes, public rental bicycles, and other types of green transportation.Public Services: information of police and firemen stations, public health organizations, hospitals, and schools, among others.

##### Big Data Context

This represents the core business logic that gathers all the contextual information of the user and uses it to assess the individual health risk and pollution exposure. This layer supports data fusion, which provides personalized interventions that are implemented considering a context for risk and resilience, which results from the local policies and directives‘ assessment on urban health and resilience [38]. The intervention is then personalized according to the available local resources in terms of educational materials, events to join, recommended places, or challenging activities to adopt specific household or healthy behaviours.

Furthermore, this layer enables data fusion, which supports the integration of data and knowledge from several sources. Three data fusion schemes are used: (i) data association, (ii) state estimation, and (iii) decision fusion. These schemes support the combination of data from multisensory environments and data modelling, and allow parameter estimation from the multiple sources by providing resources that enable multi-level process dealing with the association, correlation, and combination of data and information from single and multiple sources to complete and provide timely assessments of situations and their significance.

The big data context, presented in Figure 5, comprises three main components: storage infrastructure, modelling and predictive resources, and visualization (presentation) resources. The storage infrastructure component stores all readings in a central repository, named an “urban data lake”. Data are harvested and semantically treated to allow interoperability. 

The model and intervention resources allow data modelling and abstraction over the collected data enabling data consumption; aiming at linking health, citizens’, and environmental data to detect and predict risk factors. Spatio-temporal and risk stratification models, as well as statistical analysis (such as regression and correlation analysis), are used to personalize environmental and health risk estimations, support interpolation of data, estimate pollution concentrations on urban spaces, or enable individual assessments in these domains [39]. In this context, the models comprise the data production and fusion stage, in order to support personalized interventions and information for the app, such as air pollution level and exposure to be presented in the maps (e.g., users’ data consumption) [40]. The presentation resources push produces data to the mobile application, enabling data and information presentation, and interaction trough the mobile application modules and maps. This allows the user to input data, sense-based air quality inputs, and present data to application modules (e.g., pollution value points for the pollution exposure map).

## 4. Discussion and Conclusions

This research work shares the design and development of the PulsAir application, as part of the technological developments and big data-oriented ecosystem approach of the PULSE project. Nowadays, it is recognized that the sum of individual or household behaviours should have a substantial impact on improving the surrounding environmental and health context. However, it is difficult for citizens to relate their personal behaviour to large-scale problems such as climate change, pollution, or health. Under this context, this work gathers and shares evidence, and models perceptual sensing data, coupled with a citizen science data capturing participatory approach, to do the following: (a) analyse and model health, socio-economic, and air quality data to improve lifestyle, healthier, and greener behaviours; and (b) support further policy planning and city design by creating the urban data lake, which stores the measured pollution levels as well as data from citizens regarding their mobility (i.e., geolocation) through the city, health, and socio-economic status. Moreover, the development and deployment of PulsAir helps to better understand urban air pollution and its health impacts, as well as the participatory approach through citizen science challenges and data-driven analysis of defined areas in large cities. 

The presented mobile application is being deployed and evaluated in seven major cities, in seven different countries covering three continents (Europe, Asia, and North America) with a total of 1500 users. The PulsAir application and data ecosystem was designed under a multidomain perspective (e.g., health, social, environmental, and economic) and an iterative user centric and co-creation approach. Key stakeholders and users provided feedback to several incremental mock-up versions during the system design process, ensuring the integration of requirements from citizens belonging to all regions. This means that the PulsAir application and data ecosystem have been designed to be a tool used worldwide and particularly in cities with high levels of pollution such as Paris, Singapore, or New York. 

A layered approach for data production and consumption links the interaction between the PulsAir application and the backend processing. This generates a dynamic data ecosystem that enables citizen science data capturing, fusing these data with data captured from satellites or environmental sensors in cities and the existing legacy data, which are modelled and sent to the application for visualization and use by the citizens. Smartphones, wearables, fixed/mobile environmental/weather stations, and satellites serve as nodes for environmental, mobility, and socio-economic data capturing and sharing, while public health and urban climate local repositories provide existing targeted domain information. The system integrates the smartphone application for data gathering and sharing through data uploading into the centralized repository and data lake. As data are collected from many users in different cities, the data density increases, so modelling of the data is required to provide more accurate estimates of citizen health risks and exposure to pollution. 

The comprehensive developments of the PulsAir application and supporting data demonstrate the capacity of producing an integrated and more condensed data ecosystem. The continuous large-scale collection of citizen science data (e.g., socio-economic, health, and geolocation), data from city sensors and satellites, and related legacy data will enlarge the current data environment and provide a higher data spatial resolution. This higher density of data allows richer and more accurate big data-oriented analytics and machine learning predictive models to provide more personalized feedback and environmental-health risk information to citizens (through the application) and policymakers (through the data ecosystem). Consequently, this approach will empower users by providing more informed and data-driven feedback for decision making. The citizens can move towards healthier and greener behaviours based on the recommendations received. The policy decision makers can have more clear spatial references linked to environmental and health data to better analyse the situation in the defined areas (e.g., neighbourhoods), in order to generate spatially targeted health-oriented interventions (e.g., health policies that prevent hospitalizations or better health recommendations). The actions taken by the PHOs in defined spaces or areas in the cities can produce a direct impact on citizens’ perception, generating greater motivation and empowerment, encouraging their participation in citizen science.

The application and data ecosystem still have to be validated, but the initial deployment of the seven pilots enabled a preliminary positive feedback demonstrating that the PulsAir and data ecosystem aims are realizable. Both developments are being deployed in seven major cities, in seven different countries covering three continents (Europe, Asia, and North America) with a total of 1500 users. The evaluation will provide information regarding the level of acceptance, satisfaction, effectiveness, and user experience for further scalability and larger deployments. Moreover, further results from the PULSE pilots are needed to validate the development of the context-specific and personalized risk models. Notwithstanding, our findings and approach demonstrate a step forward for understanding the infrastructure and tools needed for the full potential of citizen science. The PulsAir and data ecosystem approach provides tangible foundational developments that empower citizens to address health problems in urban environments, while enabling scientifically sound citizen science data that could be used in defined local areas of cities where the infrastructure does not provide enough volume for environmental-health predictive analytics, and thus produce appreciable value to the participant citizens that are exposed to pollution in large cities.

The PulsAir application and data ecosystem can be seen a “toolkit” that collects and process information that is rarely integrated—health data from citizens in the same area where environmental pollution is measured or identified. This approach offers to citizens a set of specifically designed applications and data ecosystems, as well as the chance to contribute to environmental science and evidence for environmental policy in a remote manner, changing the way that environmental research, monitoring, and policymaking are carried out.

## Figures and Tables

**Figure 1 sensors-19-02940-f001:**
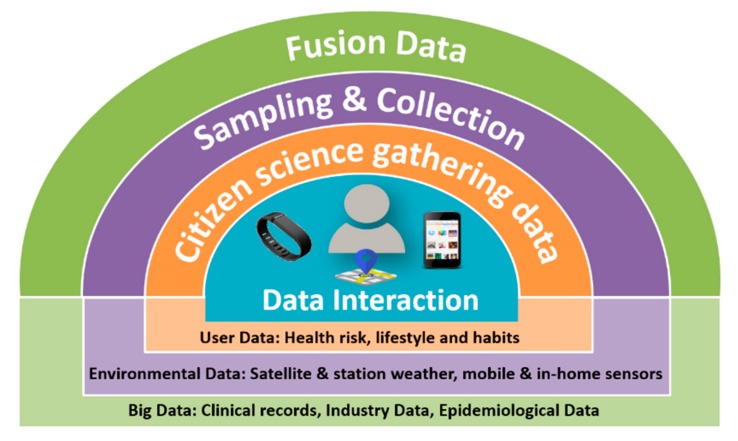
Four-step methodological approach.

**Figure 2 sensors-19-02940-f002:**
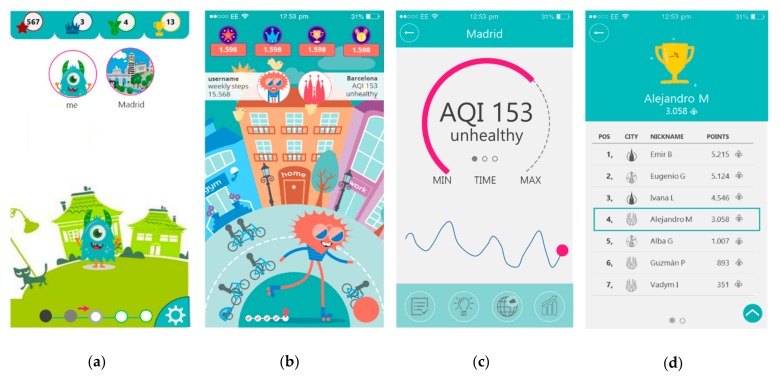
PulsAir mobile application mockup interface designs: (**a**) Home Menu, first version; (**b**) Home Menu, second version; (**c**) My City Module; (**d**) Leaderboard.

**Figure 3 sensors-19-02940-f003:**
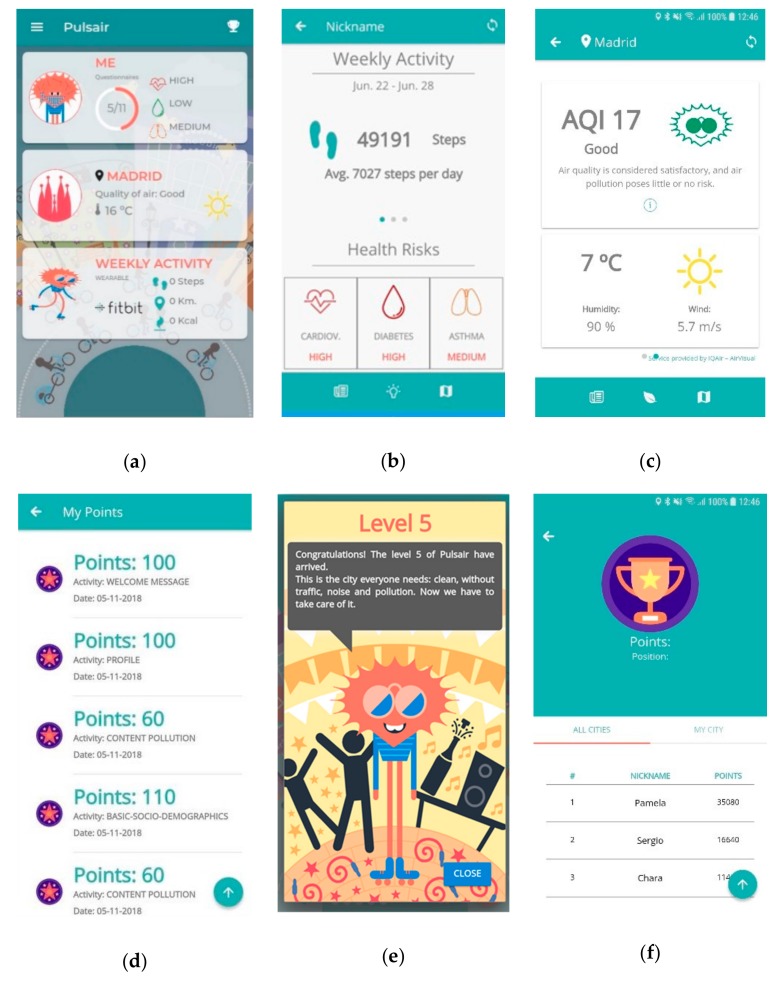
PulsAir mobile application interface screenshots: (**a**) Home Menu; (**b**) “Me” module; (**c**) “My City” module; (**d**) “My Points” section; (**e**) Level up message; (**f**) Leaderboard.

**Figure 4 sensors-19-02940-f004:**
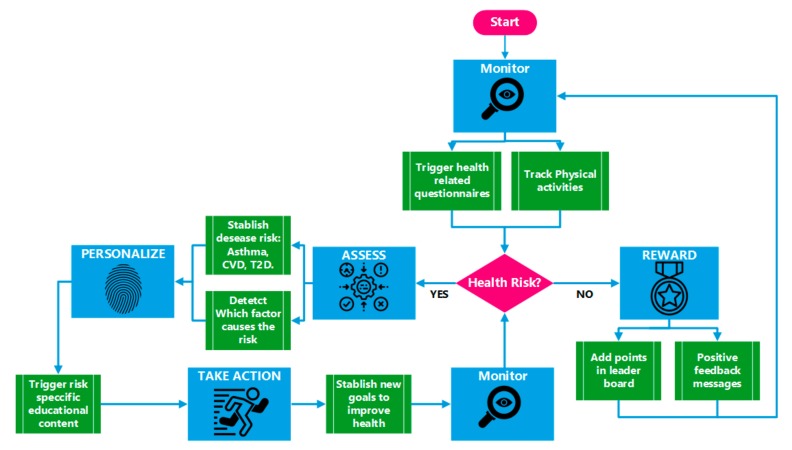
PulsAir mobile application workflow. CVD—cardiovascular disease.

**Figure 5 sensors-19-02940-f005:**
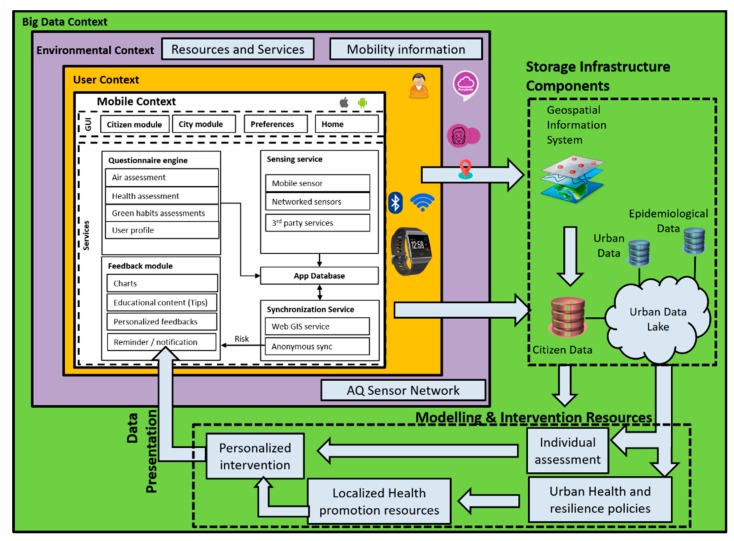
Big data-oriented ecosystem. GUI—graphical user interface.

**Table 1 sensors-19-02940-t001:** List of questionnaires.

Name	Description
Profile	Questionnaire that allows defining the socio-demographic profile of the user including age, gender, and level of education among others.
Basic Socio-Demographics	Questionnaire focused on socio-demographic status in terms of occupation, household composition, and specific questions to define if the users have been previously diagnosed with asthma or diabetes.
Neighbourhood Environment	This questionnaire seeks to study the neighbourhood of the respondents by asking questions about traffic, violence, mobility, and green areas among others [26].
Health Behaviours and Habits	Eating and smoking habits are measured with this questionnaire. Alcohol abuse is also measure based on the Alcohol Use Disorders Identification Test (AUDIT-C) questionnaire [27].
International Physical Activity (IPAQ-SF)	This questionnaire assesses the types of intensity of physical activity and sitting time that people do as part of their daily lives to estimate total physical activity [28,29].
EuroQol-5D	This questionnaire measures the health-related quality of life of the respondents through a single index value that can be used in the clinical and economic evaluation of health care and in population health surveys [30].
European Social Survey (ESS)	The ESS measures the attitudes, beliefs, and behaviour patterns of diverse populations [31].
Generalized Anxiety Disorder (GAD)	It is a tool for screening of GAD and assessing its severity in clinical practice and research [32].
Patient Health Questionnaire-9 (PHQ-9)	Used for detection of depression [33].

**Table 2 sensors-19-02940-t002:** Type of data gathered and source.

Type of Information	Scope	Data Gathering Step
Questionnaire	To gather subjective information according to the questionnaires presented in Table 1	Data interaction
GPS	To track user localization and estimate activities and cross-matching with air quality data.	Data interaction
Log App’s usage	To track system adherence and acceptance	Data interaction
Activity	To track physical activity. To monitor mobility	Citizen science data gathering
Sleep	To assess quality of sleep	Citizen science data gathering
Heart Rate	To assess cardiovascular health and fitness level	Citizen science data gathering
Air quality	To collect pollutants data	Data sampling and collection
Mobility	To provide information about public transport (i.e., bus, metro, train) or any other transport system (e.g., car sharing, bike renting) available next to the user’s location	Data sampling and collection
City services	To facilitate the list of available services in a certain location (e.g., tourism attractions, shops, restaurants, public services)	Data sampling and collection
Epidemiological	To provide the data of population health aggregated by district including demographic, disease, and mortality information	Data fusion
City	To inform about global statistics of the city (e.g., crime ratio, number of schools, universities, number of tourists/years)	Data fusion

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
