# Peer review of "Empowering Citizens through Perceptual Sensing of Urban Environmental and Health Data Following a Participative Citizen Science Approach"

_sensors, 2019, doi:10.3390/s19132940_

Round 1

Reviewer 1 Report

this paper describes the architecture of a data management system responding to air quality and human health. it is detailed in describing the system architecture and presents only limited application data.

it does have merit as a report but i find it very difficult to gain full appreciation of the work as the description is rather too complex and needs simplification. the introduction early on of a work flow description would be very useful (we see more complex system later e.g. figs 4 and 5).

there is very little "data" in the paper. i understand this is the outline of the architecture but it is developed in collaboration with end users so please describe this type of development. selection and participation criteria to give fuller appreciation of the build phase of the work.

further description of application and plans for exploitation. there is indication of wider deployment and testing- this should be explained and described more fully. - how is the deployment being facilitated? what criteria being used to test effectiveness. this is all very future focus.

also overall need to be clear on the merits of citizen science and some of the pitfalls encountered. some background review of the approach and how this can be embedded in communities. .

Author Response

Please see the attached document with a point by point response to reviewer's comments.

Reviewer 2 Report

In my opinion presented manuscript is not enough scientific soundness. It is a concept. Eventually, presented concept/model should be described with mathematical and/or automatics solution (detailed models). Further specifics of scientific significance would be necessary.In my opinion it is not a scientific research, it is realy great idea of application concerning threat to environment and human but only from the business point of view, but does not lead to the scientific conclusion.

Author Response

(The authors gave the same response as above.)

Round 2

Reviewer 1 Report

the response to comments and revisions made have improved the merit of this report. i am happy also with response to additional referees comments.

Reviewer 2 Report

I accept the answer. With such reasoning and answers on the review, I believe that this is a concept, the application, not a new science, and that was the goal.